# COVID-19 vaccine hesitancy in Malaysia: Exploring factors and identifying highly vulnerable groups

Adi Jafar[1], Ramzah Dambul[1], Ramli Dollah[2,3]*, Nordin Sakke[1], Mohammad Tahir Mapa[1], Eko Prayitno Joko[4]

1 Faculty of Social Sciences and Humanities, Geography Programme, Universiti Malaysia Sabah, Kota Kinabalu, Sabah, Malaysia, 2 Faculty of Social Sciences and Humanities, International Relations Programme, Universiti Malaysia Sabah, Kota Kinabalu, Sabah, Malaysia, 3 Asian Institute of International Affairs and Diplomacy, Universiti Utara Malaysia, Sintok, Kedah, Malaysia, 4 Faculty of Social Sciences and Humanities, History Programme, Universiti Malaysia Sabah, Kota Kinabalu, Sabah, Malaysia

☯ These authors contributed equally to this work.
* ramlid@ums.edu.my

**Data Availability Statement:** All relevant data are within the paper and its Supporting Information files.

## Abstract

Vaccine hesitancy is a global health challenge in controlling the virulence of pandemics. The prevalence of vaccine hesitancy will put highly vulnerable groups, such as the elderly or groups with pre-existing health conditions, at a higher risk, as seen with the outbreak of the pandemic Covid-19. Based on the trends of vaccine hesitancy in the state of Sabah, located in East Malaysia, this study seeks to identify several variables that contribute to vaccine hesitancy. In addition to this, this study also determines which groups are affected by vaccine hesitancy based on their demographics. This study is based on a sampling of 1,024 Sabahan population aged 18 and above through an online and face-to-face questionnaire. The raw data was analysed using the K-Means Clustering Analysis, Principal Component Analysis (PCA), Mann-Whitney U Test, Kruskal-Wallis Test, and frequency. The K-Means Clustering found that more than half of the total number of respondents (Cluster 2 = 51.9%) tend to demonstrate vaccine hesitancy. Based on the PCA analysis, six main factors were found to cause vaccine hesitancy in Sabah: confidence (var(X) = 21.6%), the influence of local authority (var(X) = 12.1%), ineffectiveness of mainstream media (var(X) = 8.4%), complacency (var(X) = 7.4%), social media (var(X) = 6.4%), and convenience issues (var(X) = 5.8%). Findings from both Mann-Whitney U and Kruskal-Wallis tests demonstrate that several factors of group demographics, such as employment status, level of education, religion, gender, and marital status, may explain the indicator of vaccine hesitancy. In particular, specific groups tend to become vaccine hesitancy such as, unemployed, self-employed, students, male, single, level of education, and Muslim. Findings from this empirical study are crucial to inform the relevant local authorities on the level of vulnerability among certain groups in facing the hazards of COVID-19. The main contribution of this study is that it seeks to analyse the factors behind vaccine hesitancy and identifies which groups more likely hesitant toward vaccines based on their demographics.

**Funding:** The author(s) received no specific
funding for this work.

**Competing interests:** The authors have declared
that no competing interests exist.

## Introduction

Every human being on this earth should be ideally spared from any form of dangerous threats.
An individual or even a group of society is very vulnerable towards various forms of hazard
[1], which can be classed into six main categories: biological hazards (biohazards), chemical
hazards, physical hazards, natural hazards, social-communicative hazards, and complex hazards [2]. Biohazards can occur either at the endemic, epidemic, or pandemic level [3]. The
novel Coronavirus or COVID-19 is one biohazard categorized as a pandemic as it has spread
throughout the entire world, including Malaysia [4]. The impact of the pandemic extends
beyond the boundaries of ecological, national, economic, and social domains [5]. The effect of
this virus increases death risks [6], psychological problems [7], and public health issues [8].
According to WHO, the global crisis caused by the COVID-19 pandemic is far more hazardous than SARS, MERS, and H1N1 [9]. In fact, the effects of this pandemic are far more devastating than geohazards such as hurricanes, earthquakes, tsunamis, and floods [10].

Therefore, several drastic measures have been taken by governments all around the world
to reduce the risk of the COVID-19 pandemic transmission. The Malaysian government, for
instance, has implemented lockdowns and the National COVID-19 Immunisation Program
(PICK) to curb the further spread of the virus [11]. Unfortunately, the implementation of lockdowns or Movement Control Order (MCO) over a long period of time in several countries
[12], including Malaysia, has caused an economic crisis, either at the individual or national
level. The rates of unemployment and loss of income were found to increase steadily, particularly among Malaysians [13]. The impact of the economic crisis has also led to more cases of
mental health problems (anxiety and depression), especially among the poor, the young,
women, and students [14]. During the implementation of lockdowns, one in three individuals
in Malaysia suffered from depression [15]. It is even more unfortunate when suicide cases
increase due to mental health issues and unemployment rates from these lockdowns [16].

This circumstance demonstrates that constant lockdowns are not the best way to combat
the spread of COVID-19. As a result, a new and more effective mechanism is required to
replace the use of lockdowns. Based on this exigency, the Malaysian government established
PICK, which serves as a coping strategy or mechanism to increase the Malaysian population's
herd immunity in dealing with the COVID-19 hazard. This initiative aims to vaccinate 80 percent (23.6 million) of Malaysia's population by February 2022 [17]. According to Kwok et al.,
85 percent of herd immunity can only be achieved after a minimum of 55 percent of the population in an area is vaccinated [18]. On the other hand, Kadkhoda opined that the chain of the
pandemic spread can only be broken if 60 to 90 percent of the population is vaccinated [19].
However, the percentage value is still influenced by the number of infection cases and vaccine
effectiveness. Hence, the practice of new norms through the implementation of lockdowns,
which partially burdens the lives of the community, can be abolished if most of the Malaysian
population has achieved herd immunity through PICK. The importance of PICK is increasingly proven when most countries make the SARS-CoV-2 vaccine their last hope to protect the
population and economy from the dangers of COVID-19 [20].

Unfortunately, until mid-2022, the number of PICK registrations (particularly for booster
dose) in Malaysia remained low and unsatisfactory [21], owing mostly to poor participation,
particularly in East Malaysia (Sabah) [22]. Among 14 states in Malaysia, Sabah recorded the
second-lowest vaccination rate, just behind Kelantan [21]. On 23 May 2022, the percentage
value of registration in Sabah was only at 64.9 percent for those who have completed 2 doses
dan 24.1 percent for those who have taken a booster dose [21]. The percentage value is still relatively low and far below the rate of global reception, which ranges from 54.8 to 88.6 percent
[23]. The situation became more complicated when many individuals failed to attend their

vaccination appointments, although they had registered for PICK [24]. Based on a review of previous studies, several factors lead to vaccine hesitancy presumably may impede the success of any vaccination program. According to MacDonald, the problem of vaccine hesitancy is usually influenced by three factors [25]. These include the issue of confidence (trust in health professionals, vaccines, and their effectiveness), complacency (quality of vaccination services, geographical accessibility and convenience of health services) and convenience (low awareness of the risk of vaccine-preventable diseases and its importance). For example, the Muslim community in Malaysia is taking the certification of vaccines' "halal" status as this seriously may affect their confidence to take the vaccine [26,27]. Similarly, Muslim communities elsewhere, such as Nigeria, Pakistan, South Africa and Afghanistan, refused to take the vaccine as there is a widespread belief that the COVID-19 will not pose any health risks [28]. In addition to the three main factors highlighted by MacDonald, social media will also influence the personal decision to take the vaccine, as illustrated in the case of Israel [29], Kuwait [30], Saudi Arabia [31] and the United Kingdom [32]. Women in these four countries were more likely to reject the vaccine due to mistrust of information from mainstream media which are usually channeled through the state-government institutions. Widespread false information about the health risks associated with taking the vaccine through social media also further adds to the public's fear as seen in Nigeria [33] and Taiwan [34].

The existing studies suggest that many key factors contribute to the prevalence of vaccine hesitancy worldwide. In this study, there is a parallel relationship between vaccine hesitancy groups and groups at higher risk of COVID-19. Ostensively, the trend of vaccine hesitancy among the local population will impede the government's effort to achieve the community's herd immunity. Based on that, this study aims to examine the factors that cause vaccine hesitancy among the people of Sabah while simultaneously identifying those who experience vaccine hesitancy.

## Materials and methods

### Model framework

The level of vulnerability is one of the essential elements that influence disaster risk in an area. This is because disaster risk will not exist without vulnerability, even in the presence of hazards. In other words, disasters will only occur when a hazard is present during a vulnerable situation [35]. The higher the level of vulnerability, the higher the level of disaster risk when a hazardous event occurs. Apart from hazards and vulnerability, the coping capacity element greatly influences the level of disaster risk. This is because coping capacity determines whether the society can withstand the disturbance of hazards or vice versa [36]. The low level of coping capacity, on the other hand, causes the level of disaster risk to increase [37]. Therefore, increasing coping capacity by empowering coping strategies or coping mechanisms is crucial in managing disaster risks [38]. The relationship between vulnerability, hazards, disaster risk and, coping capacity is shown in the following model:

$$\textit{Disaster Risk} = \frac{\textit{Hazard} \times \textit{Vulnerability}}{\textit{Coping Capacity}}$$

Source: Modified from Villagrán de León JC.Vulnerability: a conceptual and methodological review. SOURCE) Studies Of the University: Research, Counsel, Education-Publication Series of UNU-EHS, 2006, 4, 540 and Van Niekerk D. Introduction to disaster risk reduction. USAID, Washington, USA, 2011

In the context of this study, the high percentage of vaccine hesitancy in society will increase vulnerability. A high level of vaccine hesitancy will inhibit the increase in the community's

herd immunity in dealing with COVID-19 hazard. Low coping capacity will make the community more vulnerable to disaster risk (mortality, socioeconomic problems, mental health problems, and suicide).

## Data collection and procedures

This study applies the cross-sectional survey design to obtain information in the field. A total of 1,024 respondents, aged 18 years and above, have been sampled in this study. Based on the table put forward by Adam [39], with a confidence level of 99%, a total sample of 463 people is needed to represent the Sabah population of 3,904,500 people [40].

Hence, a total sample of 1,024 used in this study is sufficient to represent the study population as it has exceeded the minimum sample of 463 people. Respondents were selected using a simple random sampling technique and were asked to answer questions through an online KoBoToolbox software uploaded to WhatsApp and Facebook. This served as a safety measure to avoid exposure to the infection of the COVID-19 virus. However, the questionnaire data collection process was also conducted in several rural areas that do not have a good internet connection. This prevents those without internet access from being unrepresented in the study. The medium of instruction used in this questionnaire is in Bahasa Malaysia. The data collection process was completed in less than two weeks, from 30th of March 2021 to 15th of April 2021.

## Instrument development for questionnaires

The questionnaire in this study is divided into Sections A and B. Section A of the questionnaire focuses on respondents' socio-demographics. Section B focuses on questions regarding the respondents' perceptions toward PICK. The questions in Section A are in the form of a nominal scale, while the questions in Section B are in the form of a Likert scale with five answer choices (from 1 = strongly disagree to 5 = strongly agree). All questions in Section B are in the form of negative questions (refer to Table 1). This means that respondents who answered

**Table 1. Variables of Section B.**

| Aspect | Issues |
|---|---|
| Internal Factors | In1) Not convinced with the legality (halal) of the vaccine |
| | In2) Vaccines are not safe for my body |
| | In3) Vaccines are just a conspiracy |
| | In4) Waiting for future vaccines that should be safer |
| | In5) Not convinced that vaccines can prevent Covid-19 transmission |
| | In6) I am afraid to be injected |
| | In7) Less interested in vaccines as many recovers without vaccines |
| | In8) The practice of SOPs is sufficient to prevent the transmission of Covid-19 without vaccines |
| | In9) Still worried about being infected with Covid-19 even after being vaccinated |
| | In10) Will take vaccines only on job demands |
| External Factors | Ex1) Limited information regarding the Covid-19 immunisation program |
| | Ex2) Limited information regarding the vaccines |
| | Ex3) Vaccine-related information in the mainstream media is not convincing |
| | Ex4) Vaccine-related viral issues influenced me not to take the vaccine |
| | Ex5) Internet access prevents me from taking the vaccine |
| | Ex6) Difficult registration process for the Covid-19 immunisation program |
| | Ex7) Taking vaccines only when the family does not object |
| | Ex8) Taking vaccines only when it is compulsory |

score 1 (strongly disagree) demonstrate highly positive perceptions to PICK, while those who chose score 5 are the opposite. The questions in Section B were adapted from Fauzi et al. [41], Rumetta et al. [42], and Salam [43]. A pilot study on 30 respondents was first conducted to test the questionnaire's reliability. The Cronbach Alpha test found the alpha value of Section B (18 items) to be .900. This means that the questionnaire of this study has a reliability level in the robustness category, making it suitable for use [44].

## Data analysis

The SPSS software version 26 was employed for statistical analysis. The statistical analyses used included the K-Means Clustering, Principal Component Analysis (PCA), Mann Whitney U, Kruskal-Wallis, and Mean Score. Fig 1 presents the analysis flow of this study.

The K-Means Clustering analysis aims to group samples into two clusters: Cluster 1 and Cluster 2. This is in line with the analysis function, which produces groups of variables with a high degree of similarity within each group and a low degree of similarity between groups [45]. The formula is shown as follows:

$$J = \sum_{i=1}^{k} \sum_{j=1}^{n} \left( \|x_i - v_j\| \right)^2 = 1$$

where $\|x_i - v_j\|$ is the Euclidean distance between a point, $x_i$, and a centroid. $v_j$ is iterated over all k points in the $i^{th}$ cluster for all $n$ clusters.

Fig 2 demonstrates the Elbow and Silhouette graphs designed using machine learning analyses (Python). Both methods (Elbow & Silhouette) were utilized to determine the best number of clusters [46]. The outcome can be divided into two main clusters (Cluster 1 & Cluster 2). The data grouped according to clusters were then analysed using mean scores. This is because the population's perception towards PICK can be measured using the mean value. The higher the mean value, the more negative the respondents' perception towards PICK.

The next procedure was to perform PCA analysis using Cluster 2 data as the mean value of the overall cluster variables, higher than Cluster 1. PCA is a multivariate technique that analyses a data table describing observations by using several inter-correlated quantitative dependent variables. Its goal is to identify or extract vital information from the statistical data to represent

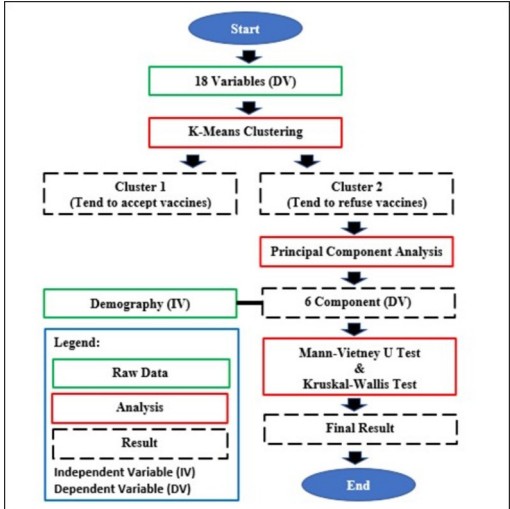

**Fig 1. Analysis flow chart.**

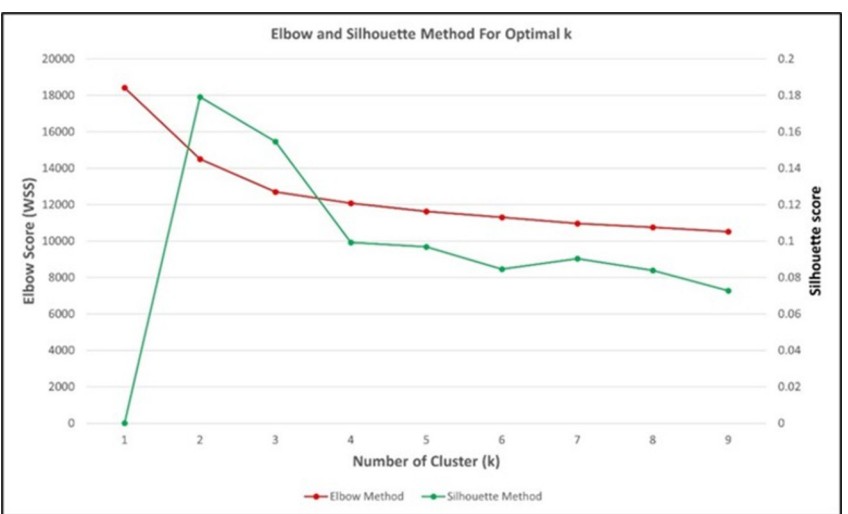

**Fig 2. Cluster number determination technique based on the Elbow and Silhouette methods.**

as a set of new orthogonal variables called principal components [47]. The PCA analysis found that the Kaiser-Meyer-Olkin (KMO) value of 0.769 belongs to the 'middling' category and is eligible for use [48]. The results of the Bartlett's test of Sphericity were also significant ($X^2$ = 2232.07, df = 153, p < 0.05), indicating that the correlation between the 18 variables is sufficient to conduct the Principal Component Analysis (PCA). The inter-correlations amongst the items are calculated, yielding a correlation matrix as shown in Table 2. Through the PCA analysis on Cluster 2, six components or main factors (Fig 3) were formed with a cumulative value of variance amounting to 61.8 percent (Table 3). A total of 61.8 percent of vaccine hesitancy factors stem from these six components, while other factors influence 38.2 percent [49]. According to

**Table 2. Correlation matrices.**

|       | In1    | In5    | In2    | In4    | Ex2    | Ex3    | Ex1    | Ex4    | Ex6    | Ex5    | In6    | In3    | In7    | In8   | In9    | Ex7    | In10  | Ex8 |
|-------|--------|--------|--------|--------|--------|--------|--------|--------|--------|--------|--------|--------|--------|-------|--------|--------|-------|-----|
| In1   | 1      |        |        |        |        |        |        |        |        |        |        |        |        |       |        |        |       |     |
| In5   | 0.593  | 1      |        |        |        |        |        |        |        |        |        |        |        |       |        |        |       |     |
| In2   | 0.551  | 0.581  | 1      |        |        |        |        |        |        |        |        |        |        |       |        |        |       |     |
| In4   | -0.154 | -0.255 | -0.189 | 1      |        |        |        |        |        |        |        |        |        |       |        |        |       |     |
| Ex2   | -0.173 | -0.101 | -0.143 | -0.149 | 1      |        |        |        |        |        |        |        |        |       |        |        |       |     |
| Ex3   | 0.104  | 0.099  | 0.09   | 0.136  | 0.285  | 1      |        |        |        |        |        |        |        |       |        |        |       |     |
| Ex1   | -0.089 | -0.104 | -0.103 | 0.171  | 0.494  | 0.327  | 1      |        |        |        |        |        |        |       |        |        |       |     |
| Ex4   | -0.127 | -0.132 | -0.079 | 0.216  | 0.061  | 0.161  | 0.122  | 1      |        |        |        |        |        |       |        |        |       |     |
| Ex6   | 0.199  | 0.138  | 0.104  | -0.052 | -0.118 | -0.138 | -0.081 | -0.152 | 1      |        |        |        |        |       |        |        |       |     |
| Ex5   | -0.205 | -0.25  | -0.244 | 0.179  | 0.025  | 0.008  | 0.057  | 0.1    | 0.069  | 1      |        |        |        |       |        |        |       |     |
| In6   | -0.147 | -0.184 | -0.08  | 0.046  | 0.098  | -0.04  | 0.071  | 0.184  | -0.089 | 0.168  | 1      |        |        |       |        |        |       |     |
| In3   | 0.201  | 0.16   | 0.103  | 0.02   | 0.052  | 0.246  | 0.064  | 0.155  | -0.061 | 0.014  | 0.151  | 1      |        |       |        |        |       |     |
| In7   | 0.195  | 0.163  | 0.204  | -0.002 | 0.006  | 0.113  | 0.063  | 0.102  | -0.073 | -0.132 | -0.039 | 0.381  | 1      |       |        |        |       |     |
| In8   | 0.034  | 0.02   | 0.024  | 0.068  | 0.015  | -0.006 | 0.075  | 0.085  | -0.117 | 0.021  | -0.03  | 0.147  | 0.382  | 1     |        |        |       |     |
| In9   | 0.43   | 0.466  | 0.546  | -0.171 | -0.163 | -0.004 | -0.131 | 0.116  | 0.18   | -0.274 | 0.133  | 0.184  | 0.312  | 0.027 | 1      |        |       |     |
| Ex7   | 0.372  | 0.339  | 0.418  | -0.126 | -0.136 | -0.01  | -0.107 | -0.119 | -0.209 | -0.224 | -0.078 | -0.055 | 0.265  | 0.047 | 0.53   | 1      |       |     |
| In10  | -0.25  | -0.243 | -0.213 | 0.446  | 0.094  | 0.031  | 0.175  | 0.198  | -0.185 | 0.12   | 0.117  | -0.118 | -0.024 | 0.041 | -0.303 | -0.237 | 1     |     |
| Ex8   | -0.231 | -0.212 | -0.156 | 0.342  | 0.026  | -0.07  | 0.083  | 0.146  | -0.101 | 0.116  | 0.105  | -0.1   | -0.045 | 0.037 | -0.168 | -0.132 | 0.542 | 1   |

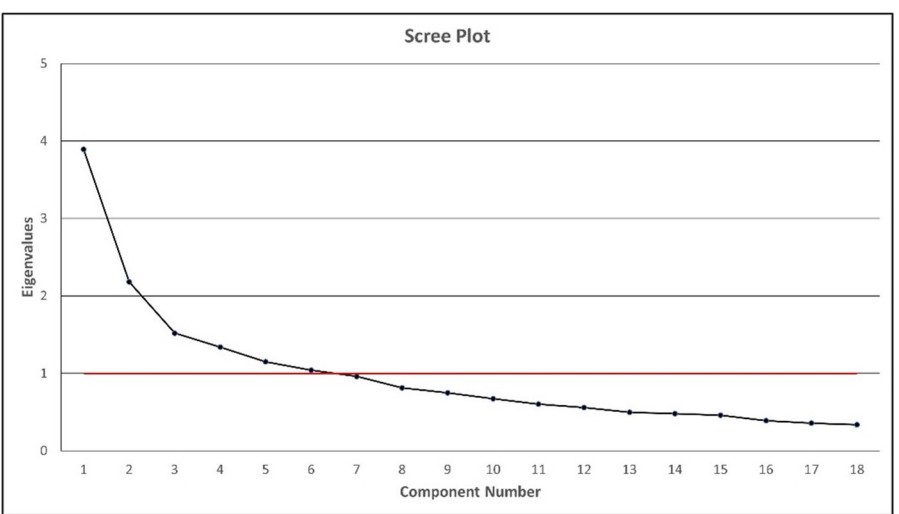

**Fig 3. Number of components.**

Hair et al. the cumulative value of the variance of more than 60 percent can be accepted. Therefore, the total variance percentage recorded in this analysis can be accepted [50].

The next step was to conduct non-parametric tests: Mann-Whitney U and Kruskal-Wallis. The Kolmogorov Smirnov normality test found that the score distribution for the six components obtained due to factor analysis (PCA) was not normally distributed. Details of the normality test results can be seen in Table 4.

The Mann-Whitney U test determines whether there are significant differences based on religious background, gender, age, education level, and marital status with six vaccine hesitancy factors. The Kruskal-Wallis test aims to verify whether significant differences are presented based on the types of occupation and amount of income associated with these six components. Both analyses were conducted to identify the most vulnerable group towards vaccine hesitancy. The Kruskal-Wallis is defined as:

$$H = \frac{12}{N(N+1)}\left[\frac{R_1^2}{n_1} + \frac{R_2^2}{n_2} + \ldots + \frac{R_k^2}{n_k}\right] - 3(N+1)$$

Where,

$R_1$ is the sum of the ranks of the $i$th sample

**Table 3. Variance and cumulative values of major components.**

| Component | Initial Eigenvalues | | |
|---|---|---|---|
| | Total (no.) | Variance (%) | Cumulative (%) |
| 1 | 3.893 | 21.6 | 21.6 |
| 2 | 2.181 | 12.1 | 33.8 |
| 3 | 1.519 | 8.4 | 42.2 |
| 4 | 1.341 | 7.4 | 49.6 |
| 5 | 1.149 | 6.4 | 56.0 |
| 6 | 1.042 | 5.8 | 61.8 |
| 7 | .961 | 5.3 | 67.2 |
| 8–18 | .812-.336 | 4.5–1.9 | 71.7–100.0 |

**Table 4. Normality test.**

| | Kolmogorov-Smirnova | | | Shapiro-Wilk | | |
|---|---|---|---|---|---|---|
| | Statistic | df | Sig. | Statistic | df | Sig. |
| Component 1 | .126 | 531 | .000 | .957 | 531 | .000 |
| Component 2 | .165 | 531 | .000 | .918 | 531 | .000 |
| Component 3 | .103 | 531 | .000 | .974 | 531 | .000 |
| Component 4 | .183 | 531 | .000 | .942 | 531 | .000 |
| Component 5 | .146 | 531 | .000 | .941 | 531 | .000 |
| Component 6 | .123 | 531 | .000 | .961 | 531 | .000 |

a. Lilliefors Significance Correction

$R^2_1$ is the sum of the ranks squared for the first sample

$R^2_2$ is the sum of the ranks squared for the second sample, and so on

$n_1$ is the number of observations in the first sample

$n_2$ is the number of observations in the second sample, and so on

$N$ is the total number of observations ($N = n_1 + n_2 + \ldots + n_n$)

$k$ is the number of populations being compared

Mathematically, the Mann Whitney U statistics are defined for each group by the following:

$$U_x = n_x n_y + \left( \frac{n_x\,(n_x + 1)}{2} \right) - R_x$$

$$U_y = n_x n_y + \left( \frac{n_y\,(n_y + 1)}{2} \right) - R_y$$

Where,

$n_x$ is the number of observations or participants in the first group

$n_y$ is the number of observations or participants in the second group

$R_x$ is the sum of the ranks assigned to the first group

$R_y$ is the sum of the ranks assigned to the second group

## Ethical considerations

The study was conducted according to the guidelines of the Ethics Committee set by the Universiti Malaysia Sabah (UMS) Review Board (Ref No UMS/FSSK6.2/100-2/2/3). After being informed about the purpose of the study and research objectives, written consent from the participant was obtained at the start of the online survey. Privacy and confidentiality were assured.

## Results

### Comparison of perceptions between Cluster 1 and Cluster 2 towards PICK

Differences of perceptions regarding PICK do exist among the population in East Malaysia. These perceptions can be generally divided into two main clusters: Cluster 1 and Cluster 2.

The K-Means analysis found that respondents in the Cluster 2 category have higher negative perceptions of PICK than Cluster 1. This was proven when the mean value of the overall variables in the Cluster 2 category was higher than Cluster 1. For the Cluster 2 category, Variable Ex6 and Variable In4 each obtained mean scores with the lowest (M = 2.35, SD = 1.086) and highest (M = 3.89, SD = .868) values. Unlike Cluster 1, Variable In5 has a mean score with the lowest value (M = 1.60, SD = .549), while Variable In4 exhibited the highest mean value (M = 3.54, SD = 1.602). The analysis results also found that more than half (In1, In2, In3, In5, In6, In7, In8, In9, Ex4, & Ex7) of the total variables had a mean value difference of more than 1, while the rest (In4, In10, Ex1, Ex2, Ex3, Ex5, Ex6, & Ex8) had a mean value difference of 1 (refer to Table 5). The study outcomes reveal that Cluster 2 respondents belong to a group of people who doubt the vaccine (vaccine hesitancy/refusal).

What is more worrying is that the total percentage of samples in the Cluster 2 category (51.9%) is higher than Cluster 1 (48.1%). In contrast, those who have not registered for PICK in the Cluster 2 category are higher (62.7%) than Cluster 1 (19.5%). In comparison to the demographic background, young people aged between 18 to 40 were more concentrated in Cluster 2 (n = 438, 82.5 percent) than Cluster 1 (n = 376, 76.3 percent). The gender composition in both clusters is not much different when the percentage of female respondents is more (C1 = 54%, C2 = 53.7%) than the percentage of male respondents (C1 = 46%, C2 = 46.3). The total percentage of single individuals in these two clusters was also higher (C1 = 54%, C2 = 61.4%) than married (C1 = 46%, C2 = 38.6%). Most respondents in these two clusters earn low income (C1 = 81.1%, C2 = 85.5%). From the religious aspect, most respondents in these two clusters are Muslim (C1 = 54.2%, C2 = 67.4%). The analysis results also found that Cluster 2 is dominated by students (30.9%) while Cluster 1 is dominated by civil servants (33.5%) (refer to Table 6).

**Table 5. Respondents' perceptions toward PICK based on clusters.**

| Aspect | Code Variables | Cluster 1 (C1) | | Cluster 2 (C2) | | The difference of Min value |
|---|---|---|---|---|---|---|
| | | Mean (M) | Std. Deviation (SD) | Mean (M) | Std. Deviation (SD) | |
| Internal | In1 | 1.56 | 0.57 | 2.67 | 1.12 | 1.11 |
| | In2 | 1.73 | 0.66 | 3.17 | 1.13 | 1.44 |
| | In3 | 1.88 | 0.87 | 3.24 | 0.98 | 1.36 |
| | In4 | 3.54 | 1.06 | 3.89 | 0.87 | 0.35 |
| | In5 | 1.60 | 0.55 | 2.84 | 1.10 | 1.24 |
| | In6 | 2.03 | 1.00 | 3.05 | 1.23 | 1.02 |
| | In7 | 2.17 | 0.88 | 3.71 | 0.89 | 1.54 |
| | In8 | 2.57 | 1.06 | 3.72 | 0.96 | 1.15 |
| | In9 | 1.70 | 0.66 | 3.19 | 1.15 | 1.49 |
| | In10 | 2.97 | 1.11 | 3.63 | 0.96 | 0.66 |
| External | Ex1 | 3.08 | 0.98 | 3.78 | 0.78 | 0.70 |
| | Ex2 | 3.29 | 0.96 | 3.88 | 0.80 | 0.59 |
| | Ex3 | 3.03 | 0.98 | 3.87 | 0.78 | 0.84 |
| | Ex4 | 2.28 | 1.06 | 3.45 | 1.02 | 1.17 |
| | Ex5 | 2.46 | 1.08 | 2.95 | 1.13 | 0.49 |
| | Ex6 | 1.74 | 0.81 | 2.35 | 1.09 | 0.61 |
| | Ex7 | 1.94 | 0.95 | 3.13 | 1.18 | 1.19 |
| | Ex8 | 2.99 | 1.11 | 3.69 | 0.98 | 0.70 |

Table 6. Demographic profile of respondents based on cluster.

| Item | Category | Cluster 1 (C1) | | Cluster 2 (C2) | |
|---|---|---|---|---|---|
| | | Number of participants | (%) | Number of participants | (%) |
| **Registration Status** | Registered | 397 | 80.5 | 198 | 37.3 |
| | Not registered | 96 | 19.5 | 333 | 62.7 |
| **Gender** | Male | 227 | 46 | 246 | 46.3 |
| | Female | 266 | 54 | 285 | 53.7 |
| **Age** | 18–40 | 376 | 76.3 | 438 | 82.5 |
| | > 41 | 117 | 23.7 | 93 | 17.5 |
| **Marital Status** | Single | 266 | 54 | 326 | 61.4 |
| | Married | 227 | 46 | 205 | 38.6 |
| **Educational status** | University | 336 | 68.2 | 310 | 58.4 |
| | High school and below | 157 | 31.8 | 221 | 41.6 |
| **Religion** | Muslim | 267 | 54.2 | 358 | 67.4 |
| | Non-Muslim | 226 | 45.8 | 173 | 32.6 |
| **Total household income** | <RM4361 (B40) | 400 | 81.1 | 454 | 85.5 |
| | RM4361-RM9619 (M40) | 77 | 15.7 | 62 | 11.7 |
| | >RM9619 (T20) | 16 | 3.2 | 15 | 2.8 |
| **Employment Status** | Civil servants | 165 | 33.5 | 83 | 15.6 |
| | Private sector employees | 80 | 16.2 | 130 | 24.5 |
| | Self-employed | 73 | 14.8 | 98 | 18.5 |
| | Not working | 56 | 11.4 | 70 | 13.2 |
| | Student | 119 | 24.1 | 150 | 30.9 |
| **Total by Cluster** | | 493 | 100 | 531 | 100 |
| **Total sample size** | | 1,024 (100) | | | |

## Factors of vaccine hesitancy in East Malaysia

Based on Table 7, it was found that the factors leading to vaccine hesitancy among the population in East Malaysia generally stem from issues of confidence (Co1), authority (Co2), weakness of mainstream media (Co3), complacency (Co4), social media (Co5), and convenience (Co6). The results of the PCA analysis revealed that each component has a varying degree of influence in sparking vaccine hesitancy problems. The confidence issue (var(X) = 21.6%) strongly influences the vaccine hesitancy problem among the six components or factors. Apart from that, authority (var(X) = 12.1%) and mainstream media (var(X) = 8.4%) issues also contribute significantly to this problem. The analysis outcomes also found that convenience (var(X) = 5.8%) and social media (var(X) = 6.4%) issues exhibit lower influence concerning vaccine hesitancy compared to the other four factors.

## Groups with high vulnerability towards vaccine hesitancy

The Mann-Whitney U test found that the respondents' level of vulnerability towards vaccine hesitancy based on the factor of authority did not show significant differences (p > 0.05) by religion, gender, age, education level, and marital status. This study also found that the respondents' level of vulnerability towards vaccine hesitancy based on the factor of confidence is significantly different (p < 0.05) by religion and education status. Here, Muslims (MR = 275.7) and low-educated (MR = 288.9) respondents were found to be less confident towards vaccines compared to non-Muslim (MR = 235.9) and high-educated (MR = 249.7) respondents. The respondents' vulnerability level based on the factor of mainstream media was also found to be significantly different (p < 0.05) by education status and gender. Highly educated (MR = 291.8) and male (MR = 281.6) respondents were found to have more distrust towards the mainstream media compared to the low-educated (MR = 229.8) and female (MR = 252.5)

**Table 7. Analysis results of Cluster 2 PCA.**

| Components (Co) | Domain | Code Variables | Loading Factor | Variance (%) | Cumulative (%) |
|---|---|---|---|---|---|
| **Co1** | Confidence | In1 | .763 | 21.6 | 21.6 |
| | | In2 | .805 | | |
| | | In5 | .761 | | |
| | | In9 | .740 | | |
| | | Ex7 | .636 | | |
| **Co2** | Authority | In4 | .683 | 12.1 | 33.8 |
| | | In10 | .809 | | |
| | | Ex8 | .798 | | |
| **Co3** | Mainstream Media | Ex1 | .784 | 8.4 | 42.2 |
| | | Ex2 | .771 | | |
| | | Ex3 | .685 | | |
| **Co4** | Complacency | In7 | .745 | 7.4 | 49.6 |
| | | In8 | .840 | | |
| **Co5** | Social Media | In3 | .600 | 6.4 | 56.0 |
| | | In6 | .671 | | |
| | | Ex4 | .585 | | |
| **C6** | Convenience | Ex5 | .590 | 5.8 | 61.8 |

respondents. For the factor of complacency, Muslim (MR = 274.6) and male (MR = 292.8) respondents were found to be more complacent compared to respondents who are non-Muslim (MR = 238.1) and female (MR = 242.9). The analysis results further revealed that respondents' level of vulnerability based on the social media factor significantly differs ($p < 0.05$) by marital status. Bachelors are more likely to be affected by social media content (MR = 278.5) compared to those who are married (MR = 246.2) (refer to Table 8).

**Table 8. The results of Mann-Whitney U test for Cluster 2.**

| Domain | Demography | | Frequency (%) | Mean Rank (MR) | P-value |
|---|---|---|---|---|---|
| **Confidence** | Religion | Muslim | 358 (67.4) | 275.7 | .005 |
| | | Non-Muslim | 173 (32.6) | 235.9 | |
| | Educational status | University | 310 (58.4) | 249.7 | .003 |
| | | Non-university | 221 (41.6) | 288.9 | |
| **Mainstream media** | Educational status | University | 310 (58.4) | 291.8 | < .001 |
| | | Non-university | 221 (41.6) | 229.8 | |
| | Gender | Male | 246 (46.1) | 281.6 | .026 |
| | | Female | 285 (53.9) | 252.5 | |
| **Complacency** | Gender | Male | 246 (46.1) | 292.8 | < .001 |
| | | Female | 285 (53.9) | 242.9 | |
| | Religion | Muslim | 358 (67.4) | 274.6 | .009 |
| | | Non-Muslim | 173 (32.6) | 238.1 | |
| **Social media** | Marital status | Bachelor | 326 (61.4) | 278.5 | .017 |
| | | Married | 205 (38.4) | 246.2 | |
| **Convenience** | Educational status | University | 310 (58.4) | 244.4 | < .001 |
| | | Non-university | 221 (41.6) | 296.3 | |

Mann-Whitney U test (p-value) at level of significance ($\alpha = 0.05$).

The results of the Kruskal-Wallis test, on the other hand, found that the respondents' level of vulnerability towards vaccine hesitancy based on the six factors did not present a significant difference by income level ($p > 0.05$). However, the opposite was found regarding the employment status category when significant differences occurred ($p < 0.05$) based on the factors of confidence, authority, mainstream media, and convenience. Respondents who are self-employed (MR = 314.9) and unemployed (MR = 318.4) are most vulnerable towards vaccine hesitancy due to confidence and convenience factors. Simultaneously, students are vulnerable to the mainstream media (MR = 288.8) and authority (MR = 299.5) factors (refer to Table 9).

It can be concluded that there are eight groups with a high level of vulnerability towards vaccine hesitancy: respondents who are Muslim, have low education, male, students, self-employed, have high education, unemployed, and single. Respondents who are Muslim (confidence & complacency), have low education (confidence & convenience), male (mainstream media & complacency), and students (mainstream media & authority) are each vulnerable to two factors. In contrast, self-employed, who have higher education, are unemployed, and have a bachelor's degree are vulnerable to one factor. Overall, respondents who are self-employed and have higher education are each susceptible to confidence and mainstream media factors, while respondents who are unemployed and single are vulnerable to convenience and social media factors (refer to Fig 4).

## Discussion

In this study, vaccine hesitant groups' composition was higher than those who accept vaccines (refer to Section 3.1). This situation is not uncommon or surprising, considering that vaccine hesitancy has been a global problem for quite some time [51]. It is estimated that the total

**Table 9. Results of Kruskal-Wallis test for Cluster 2.**

| Domain | Employment Status | Frequency (%) | Mean Rank (MR) | P-value |
|---|---|---|---|---|
| Confidence | Civil servants | 83 (15.6) | 223.5 | < .001 |
| | Private sector | 130 (24.5) | 270.7 | |
| | Self-employed | 98 (18.5) | 314.9 | |
| | Not working | 70 (13.2) | 301.7 | |
| | Student | 150 (28.2) | 236.9 | |
| Authority | Civil servants | 83 (15.6) | 230 | .008 |
| | Private sector | 130 (24.5) | 265.6 | |
| | Self-employed | 98 (18.5) | 260.9 | |
| | Not working | 70 (13.2) | 244.7 | |
| | Student | 150 (28.2) | 299.5 | |
| Mainstream media | Civil servants | 83 (15.6) | 272.9 | .025 |
| | Private sector | 130 (24.5) | 274.9 | |
| | Self-employed | 98 (18.5) | 235.4 | |
| | Not working | 70 (13.2) | 235.4 | |
| | Student | 150 (28.2) | 288.8 | |
| Convenience | Civil servants | 83 (15.6) | 261.9 | .034 |
| | Private sector | 130 (24.5) | 257.1 | |
| | Self-employed | 98 (18.5) | 266 | |
| | Unemployed | 70 (13.2) | 318.4 | |
| | Student | 150 (28.2) | 251.5 | |

Kruskal-Wallis (p-value) at level of significance (α = 0.05).

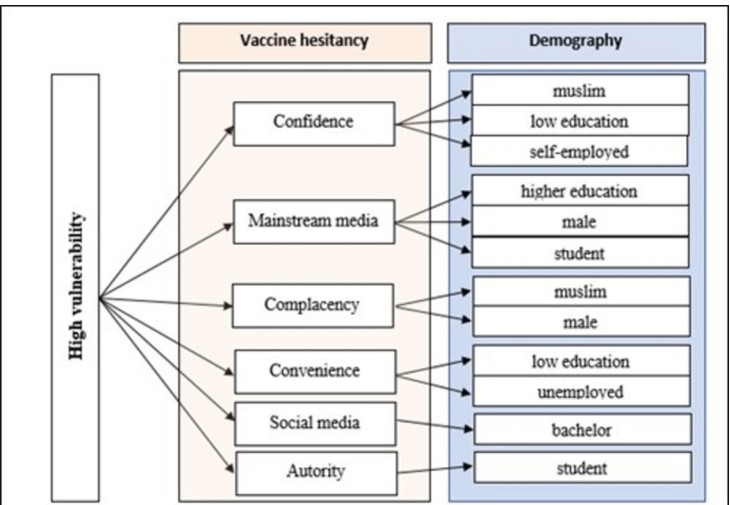

**Fig 4. Groups with a high level of vulnerability based on demographic aspects.**

percentage of vaccine hesitancy across the globe ranges from 8 to 15 percent [52]. The vaccine hesitancy problem is commonly triggered by three main factors: confidence, complacency, and convenience issues [25]. Interestingly, this study found that three other additional factors cause vaccine hesitancy aside from those raised by MacDonald. These additional factors are related to authority issues, mainstream media, and social media. In general, issues of confidence, the authorities' non-mandatory vaccination policies and mainstream media weakness are major contributors to vaccine hesitancy among East Malaysian society (refer to Section 3.2).

This study also found that the East Malaysian society's level of vulnerability towards the dangers of COVID-19 varies. It depends on an individual's demographic background. Five demographic characteristics influence level of vulnerability: religion, gender, education level, types of occupation, and marital status (refer to Section 3.3). This study did not extensively discuss the influence of age on vaccine hesitancy since no significant difference was found between young and old people in relation to this issue. However, several other studies show substantial differences between young and old people with regard to the vaccine hesitancy issue. Young people in Turkey, for instance, were found to be more likely to reject vaccines compared to the elderly [53]. The same situation applies in other developed countries like the UK and Ireland [32].

In East Malaysia, Muslims are more vulnerable to COVID-19 than non-Muslim groups. The Muslim group is more likely to experience vaccine hesitancy than non-Muslims on factors of confidence and complacency (Table 8 and Fig 4). From the aspects of confidence, the results of this study are in line with Wong [54] and Wong & Sam's [55] discovery where the vaccine's halal status is the main factor influencing the Muslims' decision on whether or not to get vaccinated. In Malaysia, the vaccine hesitancy phenomenon among Muslims is also high due to concerns over safety, in addition to the halal issue. Vaccines were alleged to contain haram substances, *mashbooh* (doubt) [56]. The assumption that vaccines are contaminated by pig DNA and have not been certified as halal worldwide further intensifies the actions of some Muslims to reject vaccines [57]. Apart from doubting the vaccine content for the Muslim community in Pakistan, the assumption that vaccines are just a conspiracy has further influenced their decision not to take the vaccine [58].

The low level of awareness (complacency) towards the importance of vaccination among Muslims in East Malaysia, on the other hand, stems from the perception that vaccines are not

necessary since many COVID-19 patients recover without being vaccinated. This group also opined that SOP practices are sufficient to curb the spread of the COVID-19 pandemic even without vaccination (Tables 1 and 7). This situation clearly shows that groups of people in East Malaysia believe that the COVID-19 pandemic does not cause serious harm to them, as thought by the Muslim community in Nigeria, Pakistan, South Africa, and Afghanistan. These groups assume that humans are designed to naturally develop immunisation against viruses without using vaccines [59]. This understanding begins from the interpretation of the Qur'an which says 'We have indeed created man in the best of molds' [60]. Hence, they assume the human body to be 'miraculous in nature and more amazing than any scientific advancement that man can achieve' [59]. In short, this proves that religious beliefs influence one's decision of whether or not to get vaccinated [52].

This study also found that men in East Malaysia are more vulnerable than women in facing the COVID-19 pandemic. This is because men are more likely to experience vaccine hesitancy due to complacency and weakness of mainstream media (Table 8 and Fig 4). This finding did not align with several previous studies. For instance, Saudi Arabian females were more exposed to vaccine hesitancy problems due to a lack of trust in government institutions, including information from mainstream media [31]. The same occurred in Israel [29], Turkey [61], Kuwait [30], and the United Kingdom [32]. In Malaysia, the lack of trust in mainstream media is not new. This is because the Malaysian public is beginning to feel that the mass media (mainstream media) has failed to effectively carry out their responsibilities as a transparent and impartial disseminator of information. The use of social media is becoming increasingly popular among Malaysians as a medium of communication in place of mainstream media [62]. Unfortunately, news from social media is more exposed to content of anti-vaccination information compared to mainstream media [62].

From the aspect of marital status, singles are more vulnerable than the married due to the tendency to experience vaccine hesitancy due to social media influence (Table 9 and Fig 4). According to Saleh & Rosli [63], social media usage among young people and students in Malaysia who are primarily single is high. Unfortunately, the anti-vaccine movement in social media, particularly in Malaysia, was reported to be actively expanding [64]. Therefore, it is not surprising if active social media users are more likely to experience vaccine hesitancy following the increasing negative prejudices against vaccination [32]. A high percentage of vaccination rejection by single or divorced people was also found in Saudi Arabia [31] and Poland [65]. In Poland, unmarried people who live alone often refuse vaccines on the grounds of waiting for the effectiveness and long-term complications of the vaccine to be assessed. In contrast, married people (with children) and those living with families are more prepared to accept vaccines because they want to protect themselves and their families from being infected by the virus.

From the aspect of employment status, on the other hand, students, self-employed, and unemployed people are the most vulnerable to the dangers of COVID-19. Students were more likely to reject vaccines due to lack of trust in information related to vaccines sourced from the mainstream media (Table 9 and Fig 4). The same situation occurred in Uganda. Medical students in the country are more interested in acquiring information on vaccines from social media and peers than traditional media (newspapers, television, radio) [66]. Saleh & Rosli found that students in Malaysia are very likely to use social media, such as Facebook and WhatsApp applications, for social purposes and to find more information [63]. Information from social media sometimes conflicts with information from the mainstream media. The discrepancy in the information obtained from both sources ends up forcing students to choose the media they feel is more authoritative. Therefore, it is not surprising if information obtained from mainstream media is less popular. The authorities in Malaysia who do not make vaccination compulsory further increase the cases of vaccine hesitancy among students (Table 9 and Fig 4).

Those who are self-employed and unemployed are more likely to reject vaccines due to factors of confidence and convenience (Table 9 and Fig 4), respectively, as had occurred in Bangladesh [67]. Compared to civil servants, both groups are less exposed to the importance of vaccination. The proof is that the government has not designed a single intensive program to increase the understanding among self-employed and unemployed people regarding the importance of vaccination. Unlike these groups, civil servants are more likely to undergo intensive training courses explicitly designed to increase their understanding of the importance of vaccination. In this course, civil servants' communication skills are further improved to disseminate effective and positive information about vaccination to the public. This is in accordance with the role of civil servants, who are described as the government's 'front-liners' [68].

The highly-educated group was found to be more likely to reject vaccines due to mistrust of the mainstream media (Table 9 and Fig 4). This is in line with the study of Tiung et al. where mainstream media does not easily influence individuals with high literacy levels [69]. This is because there is an assumption that mainstream media is a propaganda tool that is not transparent as well as biased [62]. On the other hand, the low-educated group is more likely to experience vaccine hesitancy due to convenience and confidence issues (Table 9 and Fig 4). In the context of this study, the convenience issue is related to the difficulty of registering for the vaccines. In Malaysia, the reliance on online service media is high, especially in terms of the required aspects of vaccine registration. This is evidenced when a majority of vaccine registration methods are done online, either through the MySejahtera application or the website www.vaksincovid.gov.my [17]. Unfortunately, many low-educated people, especially senior citizens, are not familiar with using online applications, especially MySejahtera [70]. This indirectly leads to cases of vaccine hesitancy among the low-educated group, which also occurred in several countries either in developed countries such as Israel [29] and the United Kingdom [61] or in developing countries such as Bangladesh [67].

## Conclusions

It can be concluded that the East Malaysian population's level of vulnerability in facing biohazards, especially the COVID-19 pandemic, varies. The demographic background of the population influences the difference in the level of vulnerability. Unemployed, self-employed, students, men, single, low education, and/or Muslims are more vulnerable to the COVID-19 virus. These groups are more likely to experience vaccine hesitancy caused by factors of confidence, mainstream media, complacency, convenience, social media, and/or authority. The existence of different vaccine hesitancy factors based on the demographic background of each group of society demands that stakeholders should be more sensitive in solving this problem. The right approach or method to address the issue of vaccine hesitancy should be adapted to the demographic background of the target group.

Empirical studies like this are beneficial as supporting sources, especially for authorities to extensively understand the society's level of vulnerability in facing the risks of COVID-19. Understanding the community's level of vulnerability based on its demographic background will facilitate the accomplishment of a more effective PICK management.

In addition, this study also has its own uniqueness compared to previous studies in discussing the phenomenon of skepticism towards the vaccine. Although the existing studies have been focused on exploring a range of factors of vaccine hesitancy, there is a limited discussion detailing the demographics of the population that will affect their decision to take the vaccine. In fact, in Malaysia, existing studies did not address the factors of demographics that may influence the trend of vaccine hesitancy. Accordingly, this study will fill the gap by identifying

at-risk groups based on their demographics. Moreover, this study is equipped with the risk assessment model as a guide to explain the relationship between factors of vaccine hesitancy and the risks associated with the loss of personal experience because of vaccine hesitancy. In addition to this, the model of risk assessment is suitable for not only assessing the geohazard risks such as floods, landslides, and global warming; but can also be applied to explore the context of biohazard studies. Nevertheless, this study also proposed several suggestions for a prospective study on vaccine hesitancy. Future studies on vaccine hesitancy should be broader and more inclusive by including larger sampling. Ideally, the sampling of the study can be explored in all states in Malaysia as this remains a gap in understanding vaccine hesitancy.

## Supporting information

**S1 File.**
(SAV)

## Author Contributions

**Conceptualization:** Adi Jafar, Ramzah Dambul, Ramli Dollah, Nordin Sakke, Mohammad Tahir Mapa, Eko Prayitno Joko.

**Data curation:** Adi Jafar, Ramzah Dambul, Ramli Dollah.

**Formal analysis:** Nordin Sakke, Mohammad Tahir Mapa, Eko Prayitno Joko.

**Investigation:** Adi Jafar, Ramzah Dambul, Ramli Dollah.

**Methodology:** Adi Jafar, Ramzah Dambul, Ramli Dollah.

**Project administration:** Nordin Sakke, Mohammad Tahir Mapa, Eko Prayitno Joko.

**Resources:** Nordin Sakke, Mohammad Tahir Mapa, Eko Prayitno Joko.

**Visualization:** Nordin Sakke, Mohammad Tahir Mapa, Eko Prayitno Joko.

**Writing – original draft:** Adi Jafar, Ramzah Dambul, Ramli Dollah.

**Writing – review & editing:** Adi Jafar, Ramzah Dambul, Ramli Dollah, Nordin Sakke, Mohammad Tahir Mapa, Eko Prayitno Joko.

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
