## [Decision Letter · Decision Letter 0]

21 Apr 2022

PONE-D-22-04772Community vulnerability in dealing with Biological Hazard (Biohazard): Overview of Covid-19 Vaccine Hesitancy phenomenon in East Malaysia (Sabah)PLOS ONE

Dear Dr. Dollah,

Thank you for submitting your manuscript to PLOS ONE. After careful consideration, we feel that it has merit but does not fully meet PLOS ONE’s publication criteria as it currently stands. Therefore, we invite you to submit a revised version of the manuscript that addresses the points raised during the review process.

ACADEMIC EDITOR: While the content of your paper is valuable and addressing an interesting and important topic, the research lacks clarity on some parts of methodology and the analysis of data.The sampling distribution have significant demographic variations in the number of registrations and could influence the results of the hypothesis. the author needs to discuss the variations in the sample and their relation to the hypothesis. Use of k-means clustering requires for equal variance of all variables and the same prior probability for all k clusters. Incase variances are unequal, you can reconsider for modified k-means.Application PCA assumes existence of correlation between the variables, so a correlation matrix and its significance values can be added prior to PCA.Authors have used non-parametric tests for testing of hypothesis. Critical tests of Normality needs to be conducted pre-testing of hypothesis and application of testing techniques.==============================

We look forward to receiving your revised manuscript.

Kind regards,

Prabhat Mittal, Ph.D.

Academic Editor

PLOS ONE

Journal Requirements:

Reviewers' comments:

Reviewer's Responses to Questions

**Comments to the Author**

1. Is the manuscript technically sound, and do the data support the conclusions?

Reviewer #1: Yes

Reviewer #2: Yes

2. Has the statistical analysis been performed appropriately and rigorously? 

Reviewer #1: Yes

Reviewer #2: Yes

3. Have the authors made all data underlying the findings in their manuscript fully available?

Reviewer #1: Yes

Reviewer #2: Yes

4. Is the manuscript presented in an intelligible fashion and written in standard English?

Reviewer #1: Yes

Reviewer #2: Yes

5. Review Comments to the Author

Reviewer #1: This article is interesting and very timely. The author has presented the paper well in accordance with scientific rules. The findings of this study are also very important for the government, especially to speed up the slow vaccination process. On the other hand, resistance to vaccination appears for various reasons. Whether it's related to insufficient information dissemination, inappropriate policies, or even the spread of hoax news related to the COVID-19 vaccine. This is actually the trigger for why the vaccination did not work as expected. The triggers for this slow vaccination process have not been seen in this paper. In fact, it is very important to know as well as an academic footing on the significance of this research. Therefore, the author is recommended to add a specific sub-section that discusses the factors of success and inhibition of vaccination which has been done by previous authors during the COVID-19 pandemic. In the introduction, the author also needs to briefly explain what community vulnerability is so that the general readers are able to grasp the theoretical meaning of this paper. This gap has implications for the discussion section where the author actually wants to refer to the previous literature but the literature review section does not yet exist. Should all of these issues be addressed in revision, I would recommend that the essay be published.

Reviewer #2: This manuscript is worth publishing, which discusses the current situation in Malaysia.

However, there are a few elements that need to improve accordingly.

Topic

- It's too long and needs to revise accurately.

Abstract

-The author needs to highlight the novelty and implication of the study properly.

Introduction

- Overall good and very clear.

- The statement of problem has been well written and straightforward. But, are the issues presented accurately in describing the current situation in 2022?

Literature Review

- Too general and only summarized in the introduction. The literature review should improve by referring to the latest studies on the subject matter, especially in 2022.

Methodology

- Very good and adequately explained.

Results

- The statistical analysis has been performed appropriately and correctly.

Discussion

- The author has successfully discussed the finding from various perspectives.

Conclusion

- The authors conclude all data underlying the findings in the manuscript but still need to highlight the novelty and implication of the study correctly.

- Need to provide suggestions from the study.

Thanks.

6. PLOS authors have the option to publish the peer review history of their article (what does this mean?). If published, this will include your full peer review and any attached files.

Reviewer #1: No

Reviewer #2: No

---

## [Author Response · Author response to Decision Letter 0]

10 Jun 2022

We have addressed all editor and reviewers’ comments and suggestions, point by point in table form as attached in the submission file

---

## [Editor Report · Decision Letter 1]

20 Jun 2022

COVID-19 Vaccine Hesitancy in Malaysia: Exploring Factors and Identifying Highly Vulnerable Groups

PONE-D-22-04772R1

Dear Dr. Dollah,

We’re pleased to inform you that your manuscript has been judged scientifically suitable for publication and will be formally accepted for publication once it meets all outstanding technical requirements.

Kind regards,

Prabhat Mittal, Ph.D.

Academic Editor

PLOS ONE

Additional Editor Comments (optional):

In my opinion, the authors have appropriately addressed the comments of the reviewer and the required changes have been made. The manuscript is addressing an interesting study on exploring vaccine hesitancy among different groups of people. The manuscript has identified the groups based on factors (demographic characteristics) that influences the vaccination drive.

Sampling framework is well presented and the descriptive statistics sufficiently displayed in a tabular manner.

Authors in the revised manuscript incorporated the correlation matrix and necessary tests of normality that completes the assumptions for implemented statistical methods.

The concerns presented in the research findings are fascinating, and have potential to aid in the creation of new approaches. In its current state, the manuscript is acceptable
---

## [Editor Report · Acceptance letter]

29 Jun 2022

PONE-D-22-04772R1 

COVID-19 Vaccine Hesitancy in Malaysia:
Exploring Factors and Identifying Highly Vulnerable Groups 

Dear Dr. Dollah:

I'm pleased to inform you that your manuscript has been deemed suitable for publication in PLOS ONE. Congratulations! Your manuscript is now with our production department. 

Kind regards, 

on behalf of

Dr. Prabhat Mittal 

Academic Editor

PLOS ONE